# Blackjack mutations improve the on-target activities of increased fidelity variants of SpCas9 with 5'G-extended sgRNAs

Péter István Kulcsár [1,2,3 ✉], András Tálas[1,4], Eszter Tóth[1], Antal Nyeste[1], Zoltán Ligeti[1,3,5], Zsombor Welker[6] & Ervin Welker [1,2 ✉]

Increased fidelity mutants of the SpCas9 nuclease constitute the most promising approach to mitigating its off-target effects. However, these variants are effective only in a restricted target space, and many of them are reported to work less efficiently when applied in clinically relevant, pre-assembled, ribonucleoprotein forms. The low tolerance to 5'-extended, 21G-sgRNAs contributes, to a great extent, to their decreased performance. Here, we report the generation of Blackjack SpCas9 variant that shows increased fidelity yet remain effective with 21G-sgRNAs. Introducing Blackjack mutations into previously reported increased fidelity variants make them effective with 21G-sgRNAs and increases their fidelity. Two "Blackjack" nucleases, eSpCas9-*plus* and SpCas9-HF1-*plus* are superior variants of eSpCas9 and SpCas9-HF1, respectively, possessing matching on-target activity and fidelity but retaining activity with 21G-sgRNAs. They facilitate the use of existing pooled sgRNA libraries with higher specificity and show similar activities whether delivered as plasmids or as pre-assembled ribonucleoproteins.

[1] Institute of Enzymology, Research Centre for Natural Sciences of the Hungarian Academy of Sciences, Budapest H-1117, Hungary. [2] Institute of Biochemistry, Biological Research Centre of the Hungarian Academy of Sciences, Szeged H-6726, Hungary. [3] Doctoral School of Multidisciplinary Medical Science, University of Szeged, H-6720 Szeged, Hungary. [4] School of Ph.D. Studies, Semmelweis University, Budapest H-1085, Hungary. [5] Gene Design Ltd, Szeged H-6726, Hungary. [6] Biospiral-2006 Ltd, Szeged H-6726, Hungary. ✉email: kulcsar.peter.istvan@ttk.mta.hu; welker.ervin@ttk.mta.hu

The *Streptococcus pyogenes* Cas9 (SpCas9) nuclease, along with other RNA-guided nucleases of the type II CRISPR system, has proved its value for genome engineering applications[1–14]. Intensive research has been focused at increasing its potential by minimizing off-target activity, which restricts its use in areas where high specificity is essential[15–23]. The most promising approaches to decrease its off-target activity are the generation of increased fidelity mutant variants, such as eSpCas9, SpCas9-HF1, and HypaSpCas9, developed by rational design[24–26], evoSpCas9 developed by exploiting a selection scheme[27] or the HeFSpCas9 variants developed by combining the mutations found in eSpCas9 and SpCas9-HF1[26,28]. Limitations of this approach include increased target selectivity, meaning that at several target sites that are otherwise cleaved by the wild-type (WT) SpCas9 these nucleases either do not cut or do so in only a limited fashion. Another limitation of using increased fidelity mutant variants is their reduced compatibility with 5′-altered sgRNAs. Indeed, most of the increased fidelity nucleases can routinely be used only with fully matching 20-nucleotide-long spacers (20G-sgRNAs)[24,25,27–30]. It is plausible that they do not work well with 5′ mismatching or truncated sgRNAs because, by design, they are inherently characterized by a lower spacer-target mismatch tolerance (i.e., they are sensitive to structural alterations within the DNA-RNA hybrid helix, which is bundled up inside the protein structure). However, it is less obvious why they possess diminished activity with 5′-extended sgRNAs, given that the extension is supposed to protrude from the structure of the nuclease[31,32]. Some of the extensions were also shown to increase the fidelity of the nuclease action, for which an explanation is still missing[20,33]. An understanding of this effect may lead to a better comprehension of the main factors that determine specificity and effectivity of the action of increased fidelity SpCas9 nucleases. This issue also has technical aspects: to comply with the sequence requirement of the promoters commonly used to transcribe the sgRNA (such as the human U6 promoter in mammalian cells[34] or the T7 promoter in vitro[35–37]), 5′ G-extended sgRNAs are frequently used with the WT SpCas9 when appropriate 20G-N19-NGG targets cannot be identified bioinformatically. Indeed, there are 27 knockout pooled sgRNA libraries at Addgene (as of 24 June 2019; https://www.addgene.org/pooled-library/) and none of them is restricted to 20G-N19-NGG target sequences. Such a shortage of appropriate targets is also a general problem with applications where there is little room to maneuver, for example when a specific position needs to be targeted by exploiting single strand oligos, when using either dCas9-FokI nucleases or base editors or when tagging proteins. Although some methods have been adapted, there is no general approach to extend the target space available for increased fidelity SpCas9 variants beyond the 20G-N19-NGG target sequences[38–41]. The use of chemically synthesized sgRNAs in pre-assembled ribonucleoprotein (RNP) form circumvents this problem in certain cases; however, RNPs are not suitable for use in pooled-library screens and are prohibitively expensive for large-scale or high-throughput studies. Furthermore, it is specifically reported that e-, -HF1, Hypa-, and evoSpCas9 have strongly reduced activities when they are applied by the RNP delivery method[42]. Other approaches exploiting ribozyme- or tRNA-sgRNA fusions have not been well characterized for the sequence dependence of sgRNA-processing. These systems have not been applied to any large-scale studies, and none of the pooled sgRNA libraries included in the 45 activation, repression or knockout libraries currently available at Addgene (https://www.addgene.org/pooled-library/) is built on ribozyme- or tRNA-sgRNA fusion vectors.

Here we report a general solution to this problem, applicable to all increased fidelity SpCas9 nucleases, that results in high specificity editing of a considerably wider target range, even when

applied as pre-assembled RNP complex, the form which can further increase editing specificity.

## Results

**Blackjack-SpCas9-HF1 works with 21G-sgRNAs.** A 5′ G-extension of sgRNAs affects the activity of the increased fidelity variants such as e-, -HF1, Hypa-, evo-, and HeFSpCas9 examined here. We proposed that it might result from a capping of the 5′ end of the sgRNA by Glu1007 and Tyr1013, which are connected via a surface loop as revealed by some X-ray structures of SpCas9 nuclease found by the Doudnas' lab and Nishimasu et al. (Fig. 1a)[43,44]. We anticipated that removing the cap by mutation would make space for a 5′ G-extension of the sgRNA without clashing with the polypeptide chain, and it could be achieved without disrupting the structural features of the folded protein. Such modification would allow the increased fidelity nucleases to work with similar efficiency when charged with sgRNAs containing either 20- or 21-nucleotide-long spacers (20G-sgRNA or 21G-sgRNA), thereby extending their target space to non-20G targets without losing fidelity. Furthermore, it has been recently reported that some 5′-extensions of the sgRNA increase the fidelity of the WT protein[20,33], which we suppose to occur mainly via the perturbation of the cap-interaction. Thus, removing the cap by mutations (i.e., perturbing the cap interaction by altering the protein instead of the 5′ end of the RNA) may also increase the fidelity of the nucleases and transform the WT protein to an increased fidelity nuclease that tolerate a 5′ extension of the sgRNA. This is a very similar rationale to that used to design SpCas9-HF1 except that in that case the interactions to be disrupted are mediated via the target DNA strand in the heteroduplex instead of the RNA strand.

We chose SpCas9-HF1 from among the high-fidelity nucleases as a starting platform and generated a mutant by replacing both Glu1007 and Tyr1013 with glycine to eliminate the presence of sidechains at these positions. In addition, we generated two deletion mutants within the region from aa. 1004 through aa. 1014. (positions where the remaining ends of the polypeptide chain seemed to be connectable without causing major distortions to the protein structure) either by completely removing this segment or by replacing it with two adjacent glycine residues, in order to eliminate the loop. Interestingly, both variants containing the deletions were active with 21G-sgRNAs (Supplementary Fig. 1a) when tested in an EGFP-disruption assay, but not the glycine-mutant. In consequence, we decided to proceed further with deletion variants. At first, we screened 13 target sites with the two deletion mutants created, along with the WT and SpCas9-HF1 (Supplementary Fig. 1b) and identified some targets that appeared suitable for easy detection of possibly improved performance of further mutant variants. Based on the same principle, we created 16 further deletion mutant candidates in that region, specifically between aa. 1003 and aa. 1017, by completely removing or exchanging segments of various lengths harboring the loop, with one to up to four amino acids with no or small side chains (Supplementary Fig. 1c). The targets selected in the previous step were used to test all mutant candidates in comparison to WT and SpCas9-HF1 (Supplementary Fig. 1d–f). The best candidate was considered to be the one exhibiting the highest on-target activity with 20G-sgRNAs and demonstrating the highest improvement with 21G-sgRNAs. This variant was named Blackjack-SpCas9-HF1 (B-SpCas9-HF1) containing only two glycine residues between the amino acids L1004 and K1014 (Fig. 1b). The Blackjack name, designated by the "B-"prefix refers to its compatibility with 21G-sgRNAs.

**Blackjack mutations increase the fidelity of the variants.** Next, we introduced the Blackjack mutations into four additional

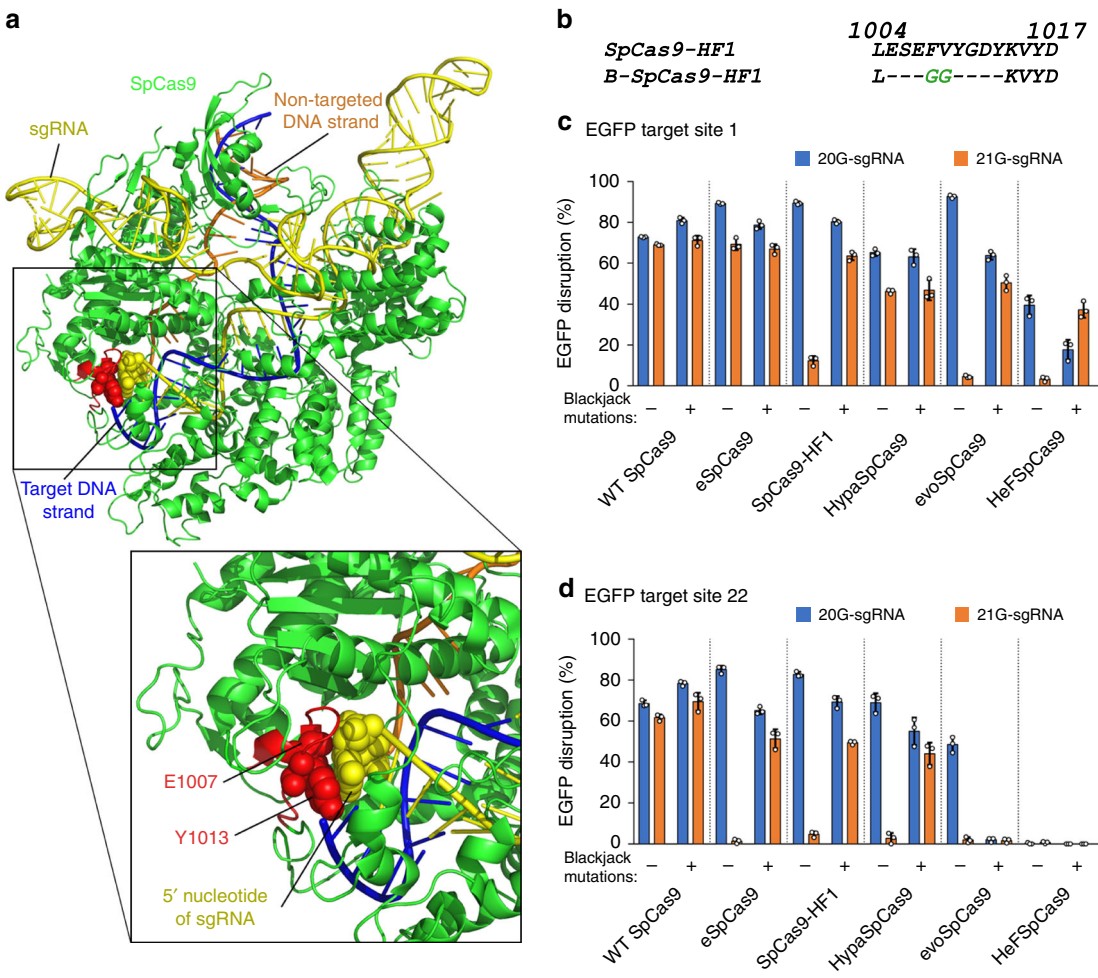

**Fig. 1 Structure-guided mutagenesis increases on-target activity of SpCas9-HF1 with 21G-sgRNAs. a** X-ray crystallography derived structure of SpCas9-sgRNA-DNA complex in the conformation closest to the cleavage-competent state (PDB ID: 5f9r)[43]. **b** Sequences of SpCas9-HF1 and the selected Blackjack-SpCas9-HF1 at the region affected, between residues L1004 and D1017; deletions (−) and insertions (green) are indicated. See also Supplementary Fig. 1. **c**, **d** Blackjack mutations increase on-target activities of increased fidelity variants with 21G-sgRNAs on different targets. Means are shown, error bars represent the standard deviation (s.d.) for $n = 3$ biologically independent samples (overlaid as white circles).

increased fidelity variants (e-, Hypa-, evo- and HeFSpCas9) and into the WT nuclease and compared the on-target activities of these nucleases with 20G- and 21G-sgRNAs to see whether they have increased activity with 21G-sgRNAs. The results obtained with two sequences are presented in Fig. 1c, d. Blackjack mutations increase the on-target activity of the variants with 21G-sgRNAs up to 17-fold; however, in case of EGFP target 22 the activity of B-evoSpCas9 decreases even with 20G-sgRNAs suggesting that Blackjack mutations may affect the activity of these nucleases on certain targets and calling for a more detailed characterization. To assess more comprehensively the effect of Blackjack mutations on the activity of increased fidelity variants with both 20G- and 21G-sgRNAs, we choose 50 EGFP targets. We found that the Blackjack mutations increase the target-selectivity (i.e., decrease the activity on certain on-target sites) of all SpCas9 variants except that of the WT (Fig. 2a, Supplementary Fig. 2). For the 21G-sgRNA experiments, to asses specifically, the effect under scrutiny, each variant pair is checked on those targets, out of the 50 where the corresponding variants with Blackjack mutations retain their on-target activities with 20G-sgRNAs. These experiments confirmed that Blackjack variants exhibit greatly increased activities with 21G-sgRNAs (Fig. 2b, Supplementary Fig. 3).

Earlier studies have shown that target selectivity and fidelity frequently increase parallel[24,25,28,45]. We were curious to find out whether the increased target selectivity caused by the Blackjack mutation also increase fidelity. Therefore, we compared their mismatch tolerance interrogating 16 out of the 50 target sequences by 144 sgRNAs, each mismatched at one position. We found that the introduction of the Blackjack mutations increases the fidelity of all SpCas9 variants (Fig. 3a and Supplementary Fig. 4a, b).

To validate these conclusions and see whether the disruption of the cap interaction increases the genome-wide fidelity of WT SpCas9 we applied GUIDE-seq analyzes on six targets. Both a 5′ G-extension and the Blackjack mutations were found to decrease the number of off-targets detected and increased the ratio of on-target vs. off-target reads compared to the WT protein. However, in case of B-SpCas9, where these capping interactions are already interrupted, the fidelity-increasing effect of the 5′ G-extension is reduced (Fig. 3b and Supplementary Fig. 5, Supplementary Data 3).

To interpret these results, we argue that the Blackjack mutations largely remove the cap from the 5′ end of the sgRNA in the cleavage-competent conformation of the SpCas9-sgRNA-DNA complex, making room for an extension and allowing the effective use of 21G-sgRNAs with increased fidelity variants.

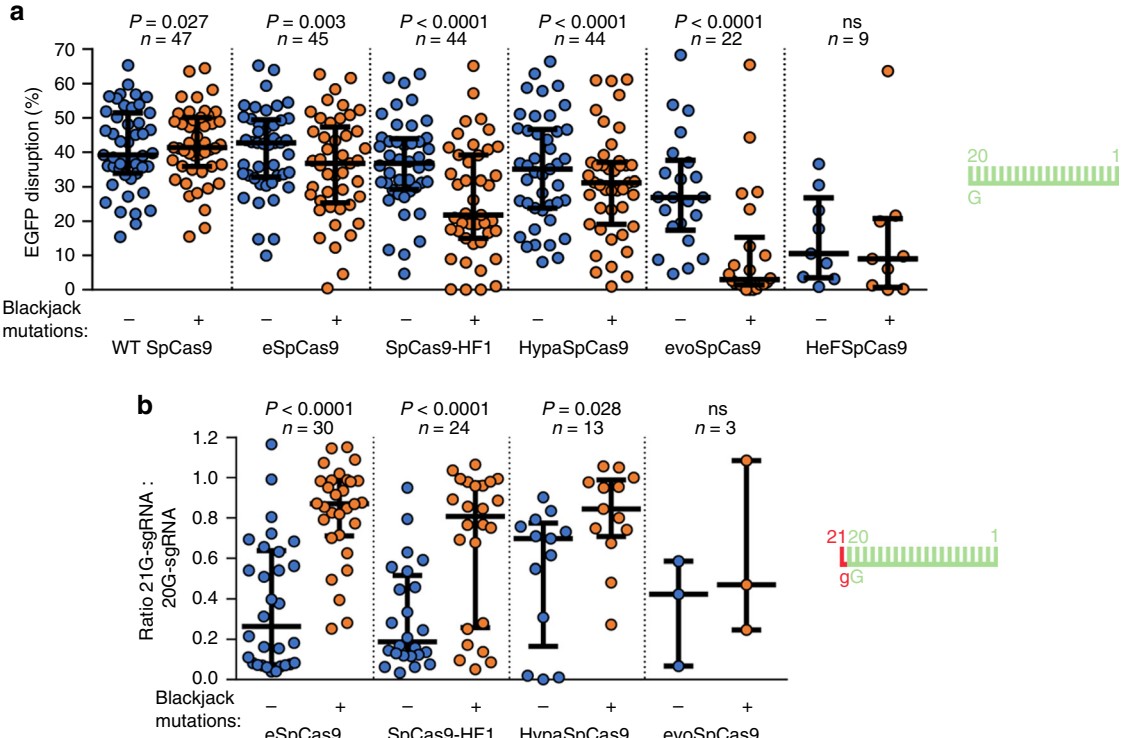

**Fig. 2 The Blackjack mutations increase not only the activity of increased fidelity nucleases charged with 21G-sgRNAs, but their target-selectivity in general. a** Blackjack mutations increase the target-selectivity of their respective parent SpCas9 variants. EGFP-disruption activities with perfectly matching 20G-sgRNAs. Results are shown only for those target sites where the SpCas9 variant without Blackjack mutations exhibits higher than background level cleavage. See also Supplementary Fig. 2. **b** On-target activities with 21G-sgRNAs on more target sites for which the SpCas9 variant with Blackjack mutations using 20G-sgRNAs exhibits at least 70% on-target activity compared to WT SpCas9. No target corresponds to this condition in the case of HeFSpCas9. See also Supplementary Fig. 3. **a, b** The median and the interquartile range are shown; data points are plotted as open circles representing the mean of biologically independent triplicates. Spacers are schematically depicted beside the charts as combs: green color teeth indicate matching-, while a red color tooth indicates the presence of an appended nucleotide within the spacer; numbering of tooth position corresponds to the distance of the nucleotide from the PAM; the starting 20th nucleotide of the spacer is indicated by an uppercase letter and an appended 21st nucleotide by a red lowercase letter. Statistical significance was assessed using two-sided Paired-samples Student's *t*-test or two-sided Wilcoxon signed ranks test as appropriate; ns not significant. A summary of data distributions and statistical details is reported in Supplementary Data 6.

However, the mutations also increase the fidelity/target-selectivity of the variants arguing for the fidelity-increasing effect of the disruption of cap interactions.

Since the effects of being able to use 21G-sgRNAs and the increased target-selectivity of Blackjack variants are confounded, and because the fidelities of the six new Blackjack variants are different from the pre-existing nuclease variants, the higher specificity editing that the Blackjack variants offer is hard to discern.

**The plus variants of e-SpCas9 and SpCas9-HF1.** To have more complete target coverage and to test our interpretation, we decided to create Blackjack variants that have identical fidelity/target-selectivity to those of eSpCas9 and SpCas9-HF1 based on the following rationale. The Blackjack mutations have two effects: The first is that the deletion potentiates cleavage with a 5′ extended 21G-sgRNA. The second is that it increases the fidelity of SpCas9 when it acts with either 20G- or 21G-sgRNAs. We proposed that by restoring some of the mutations of the Blackjack variants that originate from their corresponding parent increased fidelity nuclease to their WT residue, we can selectively compensate for the second effect. The "parental" eSpCas9 possesses three mutations (K848A, K1003A, R1060A), while SpCas9-HF1 possesses four (N497A, R661A, Q695A, Q926A). After examining the data in the studies describing their development[24,25], we constructed four and seven candidates from Blackjack-eSpCas9 and Blackjack-SpCas9-HF1, respectively, lacking one or two

"original parental" mutations at a time (Fig. 4a, d). We selected those residues for which we conjectured that their contributions to increasing the fidelity of the respective nuclease would be comparable to those of the Blackjack mutations.

For testing the residue-reverted variants in the case of SpCas9-HF1 we picked five targets on which it has considerable activity employing 20G-sgRNAs but on which B-SpCas9-HF1 exhibits strongly decreased activity with 21G-sgRNAs due to its increased target-selectivity. All new candidates exhibit increased on-target activity with 20G-sgRNAs compared to B-SpCas9-HF1 except that in which A497 was reverted and that, surprisingly, seems to show decreased on-target activities (Fig. 4b). This suggests that the target-selectivity obtained with revertants was successfully reduced and their fidelity correspondingly lowered compared to B-SpCas9-HF1 except for the A497N reversion. To find the variant whose fidelity most closely matches that of SpCas9-HF1 we employed mismatching sgRNAs to two selected targets: one for which SpCas9-HF1 exhibits close to optimal specificity and another for which it demonstrates considerable but decreased off-target activity compared to the WT nuclease. The reversion of A497 increased the fidelity of B-SpCas9-HF1 consistent with the on-target activity results. Five reversion variants lowered the fidelity of these variants below that of SpCas9-HF1, while the reversion of A661 resulted in a similar fidelity (Fig. 4c). We named this Blackjack variant as *SpCas9-HF1-plus* and selected it for a more detailed characterization.

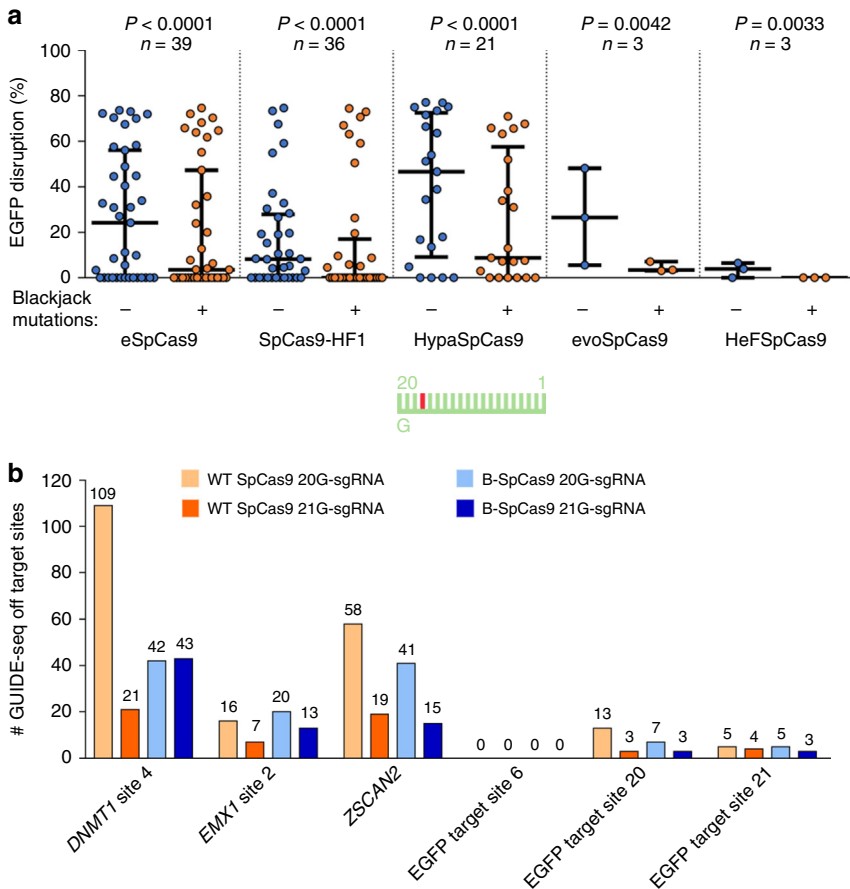

**Fig. 3 The Blackjack mutations increase the fidelity of increased fidelity nucleases. a** Blackjack mutations increase the fidelity of their respective parent SpCas9 variants. EGFP-disruption activities with partially mismatching 20G-sgRNAs. Results are shown only for those target sites where both the non-Blackjack parent- and Blackjack-SpCas9 variant exhibit at least 70% on-target activity (with perfectly matching 20G-sgRNAs) compared to WT SpCas9. Only one target (with three mismatched positions) matches this condition in the case of evo- or HeFSpCas9. The median and the interquartile range are shown; data points are plotted as open circles representing the mean of biologically independent triplicates. Spacers are schematically depicted beside the charts as combs: green color teeth indicate matching-, while a red color tooth indicates the presence of a mismatching nucleotide (not necessarily the exact position) within the spacer; numbering of the tooth positions corresponds to the distance of the nucleotide from the PAM; the starting 20th nucleotide of the spacer is indicated by an uppercase letter. Statistical significance was assessed using two-sided Paired-samples Student's *t*-test or two-sided Wilcoxon signed ranks test as appropriate; ns not significant. Summary of data distributions and statistical details are reported in Supplementary Data 6. See also Supplementary Fig. 4. **b** Bar chart of the total number of off-target sites detected by GUIDE-seq for WT and B-SpCas9 variants on six target sites targeted with 20G- or 21G-sgRNAs. See also Supplementary Fig. 5.

For B-eSpCas9, we proceeded as with B-SpCas9-HF1 to create the revertants and picked five targets using a similar rationale. Testing the residue-reverted candidate variants, all candidates showed increased on-target activities with 20G-sgRNAs on these targets (Fig. 4e). To find the variant that most closely matches the fidelity of eSpCas9 we selected two targets for which eSpCas9 exhibits close to optimal specificity but on which the B-eSpCas9 demonstrated decreased on-target activity. We employed mismatching sgRNAs to these two targets and tested the variants: the reversion of A1003 resulted in the closest fidelity match to that of eSpCas9's (Fig. 4f). We named this Blackjack variant *eSpCas9-plus* and selected it for a more detailed characterization.

Western blotting indicated that Blackjack mutations do not alter the expression level of SpCas9 variants and the amounts of the *plus* variants expressed in the steady state are comparable to those of their parent variants (Supplementary Fig. 6a). We compared the selected *plus* variants' on-target activities with 20G-sgRNAs on 25 targets with their parental variants. Both eSpCas9-*plus* and SpCas9-HF1-*plus* reached the on-target activities of their original counterpart variant on this set of target sequences (Fig. 5a and Supplementary Fig. 6b). To challenge the *plus*

variants when checking their activities with 21G-sgRNAs, different sets of ten targets were assayed with the enhanced and with the high-fidelity variants to exploit targets on which the parent nucleases exhibited strongly decreased on-target activity upon appending a 5′ 21st G. To assess specifically the effect under scrutiny, the same ten sequences were targeted with both 20G- and 21G-sgRNAs. With 21G-sgRNAs both eSpCas9-*plus* (Fig. 5b and Supplementary Fig. 6c) and SpCas9-HF1-*plus* (Fig. 5c and Supplementary Fig. 6d) demonstrated highly increased on-target activities; reaching 90% of that with 20G-sgRNAs on the same targets, in contrast to their parent variants demonstrating only 10 and 16%, respectively. To compare the fidelity of the *plus* variants with their parents, 13 targets were selected on which both eSpCas9 and SpCas9-HF1 had demonstrated reasonable on-target activities and 117 mismatching sgRNAs were employed. At all of the 39 positions examined the off-target activity of eSpCas9-*plus* resulted in an identical off-target-cleavage pattern, matching the fidelity of eSpCas9 (Supplementary Fig. 6e). The off-target activity of SpCas9-HF1 and SpCas9-HF1-*plus* compared in a similar way also gave rise to very similar patterns, closely matching each other's fidelities (Supplementary Fig. 6f). These data demonstrate

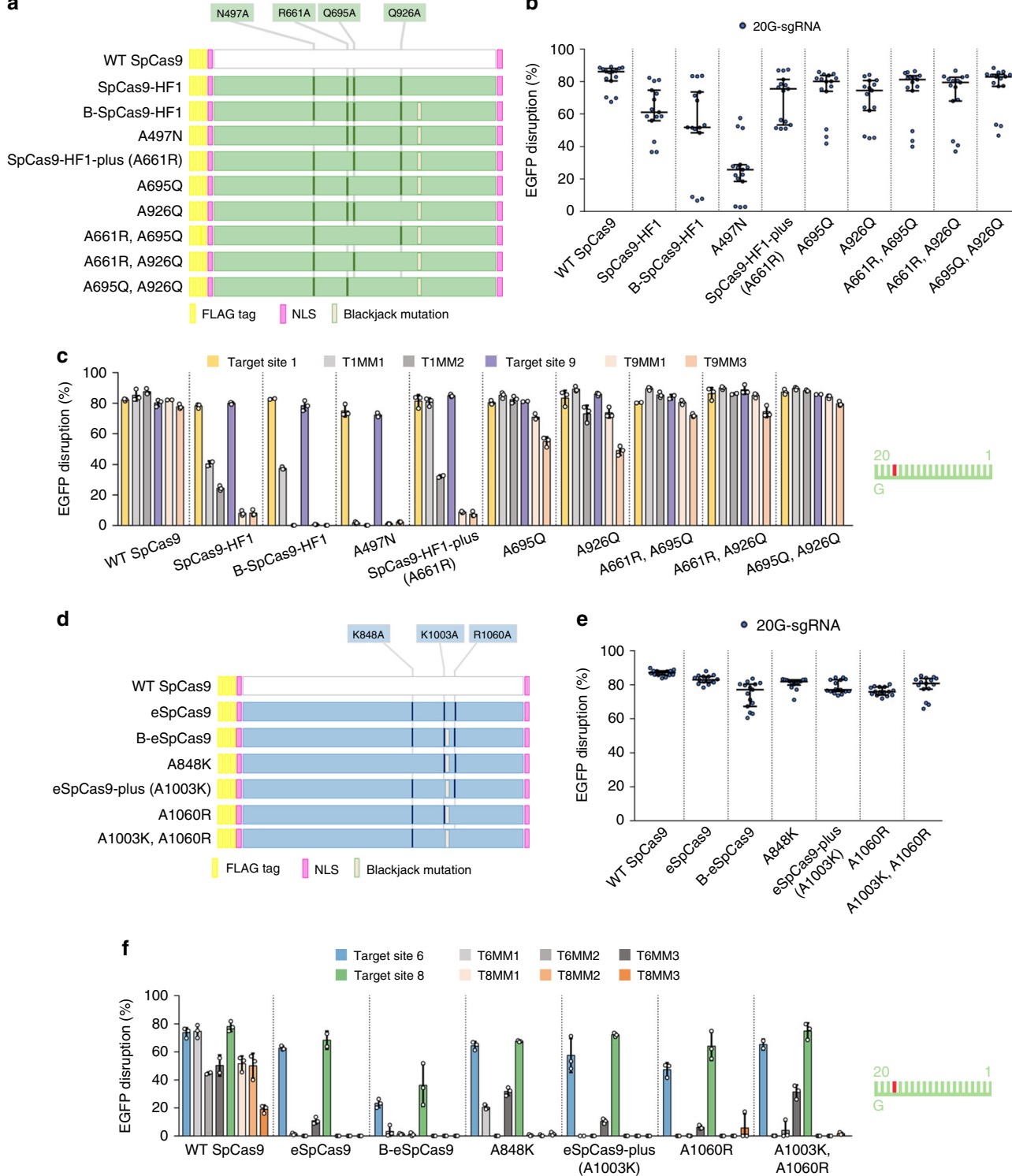

**Fig. 4 Restoring mutations to WT amino acids lowers the (on-)target-selectivity and fidelity of B-eSpCas9 and B-SpCas9-HF1. a**, **d** Schematic representation of the mutations in each variant examined. **b**, **e** On-target activities using 20G-sgRNAs measured on five target sites ($n = 3$ biologically independent samples [overlaid as blue circles]), employing the EGFP-disruption assay, median and interquartile range are shown. **c**, **f** Mismatch screen results from EGFP-disruption assay. Target sites and matching (e.g., T1, T6) or mismatching sgRNAs (e.g., T1MM1, T6MM1) are the same as in Supplementary Fig. 3. Spacers are schematically depicted beside the charts as combs: green color teeth indicate matching-, while a red color tooth indicates the presence of a mismatching nucleotide (not necessarily the exact position) within the spacer; numbering of the tooth positions corresponds to the distance of the nucleotide from the PAM; the starting 20th nucleotide of the spacer is indicated by an uppercase letter. Means are shown, error bars represent the standard deviation (s.d.) for $n = 3$ biologically independent samples (overlaid as white circles).

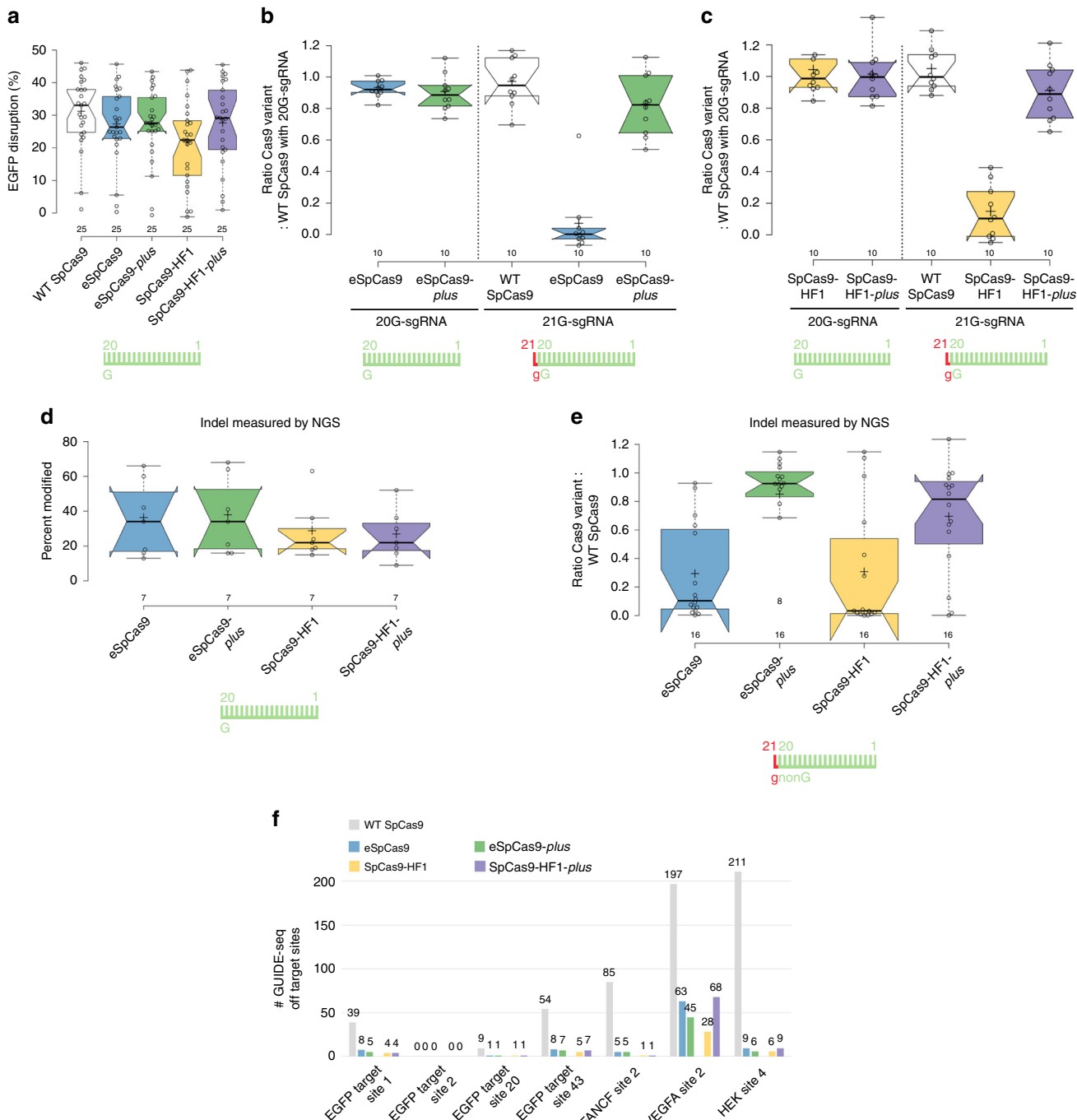

**Fig. 5 eSpCas9-*plus* and SpCas9-HF1-*plus* show greatly enhanced on-target activity with 21G-sgRNAs and identical fidelity/target-selectivity compared to eSpCas9 and SpCas9-HF1, respectively, as assessed by EGFP disruption, indels measured by NGS and by GUIDE-seq.** **a**–**c** EGFP-disruption activity **a** with 20G-sgRNAs targeting 25 sites; **b**, **c** with either 20G- or 21G-sgRNA pairs targeting two alternative sets of 10 different sequences shown as the ratio of variant activity to WT activity. **d**, **e** On-target activities of SpCas9 variants across 23 endogenous target sites within the human *VEGFA* or *FANCF* loci targeted with **d** 20G- or **e** 21G-sgRNAs, measured by amplicon resequencing. **f** Bar chart of the total number of off-target sites detected by GUIDE-seq for SpCas9 variants on seven sites targeted with 20G-sgRNAs. **a**–**e** Tukey-type boxplots by BoxPlotR[60]: center lines show the medians; box limits indicate the 25th and 75th percentiles; whiskers extend to the "minimum" and "maximum" data situated within 1.5 times the interquartile range from the 25th and 75th percentiles, respectively; notches indicate the 95% confidence intervals for the medians; crosses represent sample means; data points are plotted as open circles representing the mean of biologically independent triplicates. Spacers are schematically depicted beside the charts as combs: green color teeth indicate matching-, while a red color tooth indicates the presence of an appended nucleotide within the spacer; numbering of tooth position corresponds to the distance of the nucleotide from the PAM; the starting 20th nucleotide of the spacer is indicated by an uppercase letter and an appended 21st nucleotide by a red lowercase letter. See also Supplementary Figs. 6 and 7.

that the *plus* variants possess identical fidelity but strongly increased on-target activity with 21G-sgRNAs, and thus can be utilized on a much wider range of targets compared to their parent variants.

**NGS and GUIDE-seq characterization of the *plus* variants**. We further characterized the on-target activities of the *plus* variants in HEK293 cells, monitoring their indel-inducing activities by NGS. We selected 23 sequences from the human *FANCF* and *VEGFA* loci. There was no prior cleavage information available for them except for one (*FANCF* site 2). Sixteen of them can be targeted with 21G-sgRNAs and seven with 20G-sgRNAs (Supplementary Data 1: Target sites). The on-target activities of the *plus* variants with 20G-sgRNAs match those of their corresponding original counterparts (Fig. 5d). However, with 21G-sgRNAs they show much higher on-target activity (Fig. 5e). Since the target-selectivity of eSpCas9 and SpCas9-HF1 is higher than that of WT SpCas9[24,25,28], they are not expected to reach the 21G-sgRNAs WT-level activity. The original counterparts, eSpCas9 and SpCas9-HF1, exhibit slightly decreased on-target activities with 20G-sgRNAs, 93 and 82% on average, respectively, relative to WT SpCas9. This relative activity level must be reached by the *plus* variants with 21G-sgRNAs to allow us to say that they work with the same efficiency using either 20G- or 21G-sgRNAs. What we found was that with 21G-sgRNAs eSpCas9-*plus* demonstrates 88%, while SpCas9-HF1-*plus* exhibits 82% of the value of WT SpCas9 with 21G-sgRNA, respectively (Fig. 5e and Supplementary Fig. 6g, Supplementary Data 2) so we can say they worked with nearly identical efficiency.

We also wanted to compare the fidelity of these nucleases by GUIDE-seq. In HEK293.EGFP cells, we selected seven target sites that can be targeted by 20G-sgRNAs to make sure that not just the nuclease variants containing Blackjack mutations are able to cleave the on-target sequences. Among these targets, three (*VEGFA* site 2, HEK site 4, *FANCF* site 2) were used to characterize the off-target activities of the increased fidelity variants in earlier studies[25–27,45]. As expected, we found that all four increased fidelity variants demonstrate greatly increased fidelity compared to the WT protein and that the corresponding parent—*plus* "variant pairs" behave similarly to each other, eSpCas9-*plus* cleaving slightly less, while SpCas9-HF1-*plus* slightly more off-target sites on some targets compared to their parental variants (Fig. 5f and Supplementary Fig. 7, Supplementary Data 3). These results are consistent with the contention that the fidelity of these *plus* variants closely matches that of their parental (non-Blackjack) counterparts.

**The plus variants work effectively in RNP form**. The development of two new increased fidelity variants, Sniper and HiFi SpCas9 has been reported more recently, claiming they work effectively in RNP form, and Sniper SpCas9 is able to work even with 5′-modified sgRNAs, unlike former increased fidelity variants[42,45]. Sniper SpCas9 being less "attenuated"[45] has lower target selectivity and fidelity (Supplementary Fig. 8a, b) that may offer an explanation for its ability to work with 5′-modified sgRNAs. However, it is not clear why they possess high activity in RNP form, while the former increased fidelity variants have reported to possess a strongly reduced activity in RNP form[42]. RNPs are the method of choice for prospective clinical applications, and we wondered if Blackjack variants are able to provide optimal high-fidelity editing for the majority of the targets on which one of the other increased fidelity nucleases provide better specificity editing compared to Sniper or HiFi SpCas9. Thus, we selected 31 sequences to assay for EGFP disruption by eSpCas9 and SpCas9-HF1 and by their *plus* variants delivered in RNP

form. Since it seems that there is no consensus about the requirement of the T7 polymerase for the preferred starting sequences of the transcript, we selected sequences that start with non-G, G or GG nucleotides and we targeted them systematically with in vitro transcribed, 5′ G- or GG-extended or fully matching 20G-sgRNAs, as depicted in Fig. 6a–c. Surprisingly, in contrast to what was reported by Vakulskas et al.[42], all variants show similarly high activity with 20G-sgRNAs to that of the WT protein (Fig. 6b). There are several differences between the two experiments that could be responsible for this disagreement. The Vakulskas et al. performed several of their tests on endogenous gene targets, including on HPRT, which may behave differently than the targets in an EGFP-disruption assay, due to the chromatin context. Furthermore, here we used targets on which the activity of eSpCas9 and SpCas9-HF1 had been confirmed when introduced as plasmids, which might also account for the discrepancies with the Vakulskas study. In contrast, only the *plus* variants demonstrate high activities with 21G-sgRNAs in preassembled RNP form too, reaching up to 23-fold higher activities than their parental variants (Fig. 6a–c, Supplementary Fig. 8c–e). Thus, we conclude that *plus* variants are effective in the RNP form, provide high-fidelity editing with both 20G- and 21G-sgRNAs and allow the effective use of in vitro transcribed 21G-sgRNAs.

We were also curious to compare the on-target activities of the *plus* variant with sgRNA-processing approaches to see whether the *plus* variant offers advantages in those applications where the sgRNA-processing approach can also be applied. Testing on 19 targets, the *plus* variant showed higher activities than either the tRNA[46] or ribozyme processing approaches[29,47] (Supplementary Fig. 8f). Using both approaches, some target sequences were cleaved with reduced efficiency by the WT protein, suggesting that the understanding of the sequence dependencies of the tRNA or ribozyme processing needs a more comprehensive investigation.

We wanted to test the usefulness of Blackjack variants in a practical application by investigating the expression of the prion protein family-member Shadoo protein after inserting the EGFP sequences downstream of the mouse Shadoo promoter exploiting NHEJ repair[48]. Five NGG PAM sequences are available at relevant positions but none of them are targetable with 20G-sgRNAs, presenting a good example of where Blackjack variants can offer a specific advantage for a project. We pre-screened the available five targets with several increased fidelity nucleases by integrating an EGFP cassette into the targeted site. Pre-screening identified the optimal targets and nuclease variants (Fig. 7a) for the generation of the desired transgenic lines by increased fidelity nucleases (Fig. 7b). The results confirmed the advantage of the Blackjack variants for this practical application and we predict that comparable results will be obtained from a wide range of practical applications.

Finally, to demonstrate that the *plus* variants are compatible with pooled sgRNA library screens where the sgRNAs are inherently expressed at low levels, we generated three cell lines each expressing one 21G-sgRNA from an integrated lentivirus copy. Each of the three cell lines were interrogated by eSpCas9-*plus* in parallel to WT and eSpCas9 (Fig. 7c). In contrast to eSpCas9, eSpCas9-plus demonstrated activities approaching nearing that of the WT protein. These results confirm that the activity of eSpCas9-plus renders the nuclease compatible with pooled sgRNA libraries where the sgRNAs are naturally expressed from an integrated single copy of a lentivirus and even when 21G-sgRNAs are used.

## Discussion

Among the Blackjack nucleases developed here, three variants are likely to gain general application, replacing the corresponding

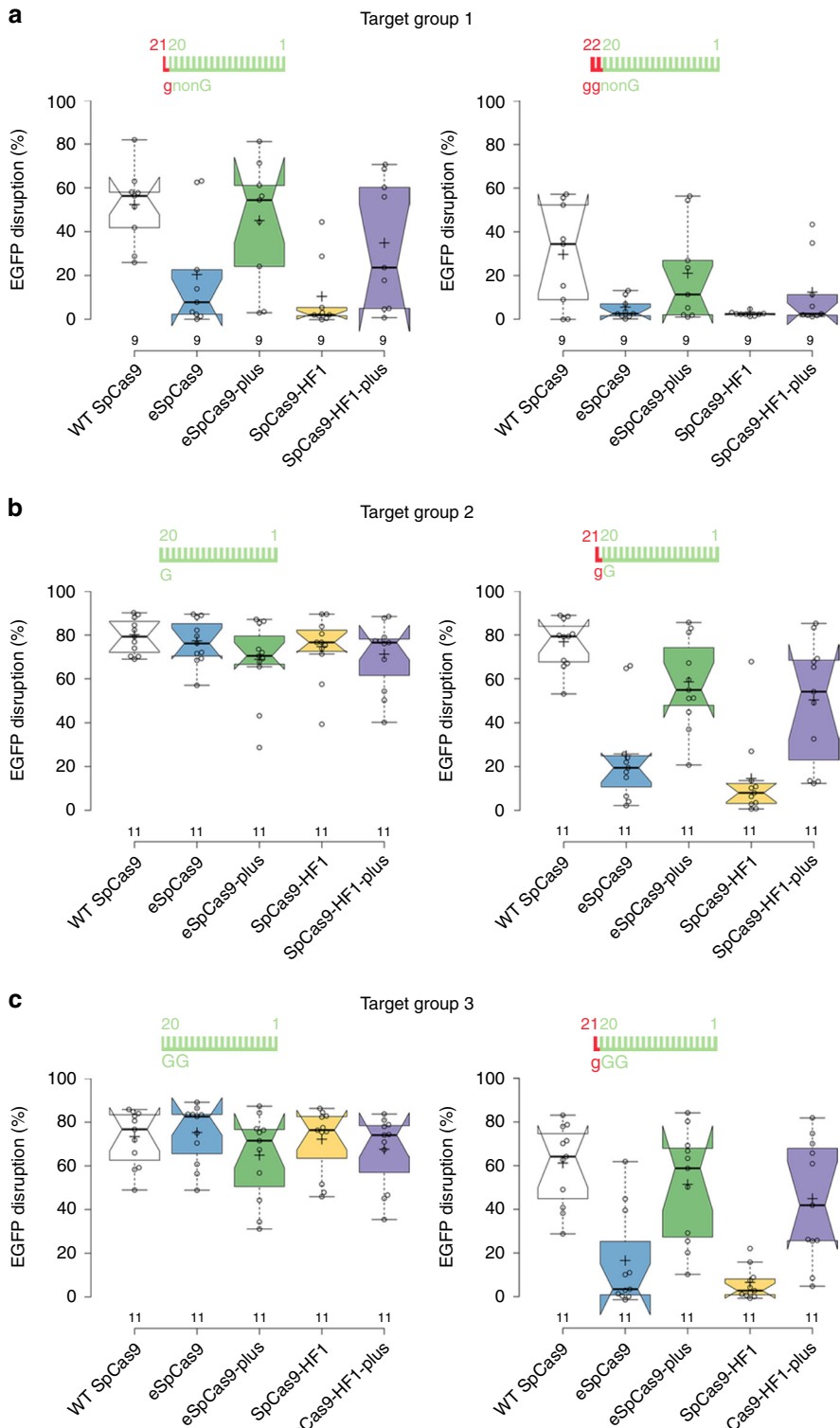

**Fig. 6 The *plus* variants are effective when transfected in pre-assembled RNP form. a–c** EGFP-disruption assays. Target sequences start with 5′ non-G-, G- or GG-nucleotides. Data from the individual samples are detailed in Supplementary Fig. 8c–e. Tukey-type boxplots by BoxPlotR[60]: center lines show the medians; box limits indicate the 25th and 75th percentiles; whiskers extend to the "minimum" and "maximum" data situated within 1.5 times the interquartile range from the 25th and 75th percentiles, respectively; notches indicate the 95% confidence intervals for the medians; crosses represent sample means; data points are plotted as open circles representing the mean of biologically independent triplicates. Spacers are schematically depicted beside the charts as combs: green color teeth indicate matching-, while a red color tooth indicates the presence of an appended nucleotide within the spacer; numbering of tooth position corresponds to the distance of the nucleotide from the PAM; the starting 20th nucleotide or dinucleotide of the spacer is indicated by an uppercase letter and an appended 21st and 22nd nucleotides by red lowercase letters. See also Supplementary Fig. 8.

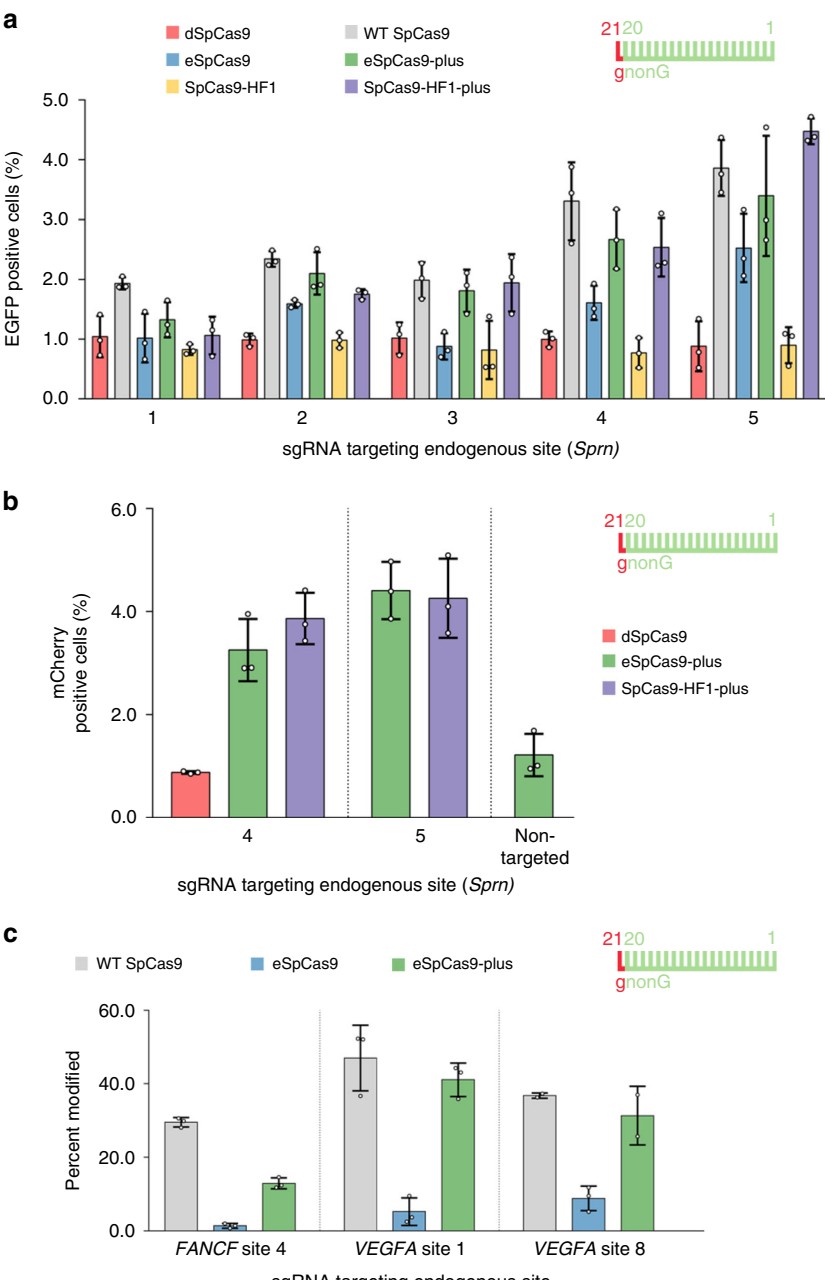

**Fig. 7 Blackjack variants facilitate modification of the endogenous Sprn gene at the 5′ coding region and are effective with sgRNAs expressed from a single-copy lentivirus. a** Pre-screening targets with increased fidelity nucleases for efficiency by the integration of a donor EGFP cassette. **b** Based on **a** the SpCas9-plus variants were selected to generate transgenic lines using the 'self-cleaving' EGFP-expression plasmid, which must integrate in-frame for *Sprn* promoter driven EGFP expression, and which downstream from the EGFP coding sequence it also contains a CMV-mCherry cassette; mCherry positive cells were counted. **c** Indel formation activity of eSpCas9-plus compared to WT and eSpCas9 with 21G-sgRNAs transcribed from an integrated single copy of lentiviruses measured by TIDE. **a–c** Means are shown, error bars represent the standard deviation (s.d.) for $n = 3$ biologically independent samples (overlaid as white circles). In the case of *VEGFA* site 8 targeted with WT and eSpCas9-plus on **c** one sample point is missing due to sample loss. Spacers are schematically depicted beside the charts as combs: green color teeth indicate matching-, while a red color tooth indicates the presence of an appended nucleotide within the spacer; numbering of tooth position corresponds to the distance of the nucleotide from the PAM; the starting 20th nucleotide of the spacer is indicated by an uppercase letter and an appended 21st nucleotide by a red lowercase letter.

non-Blackjack nucleases: eSpCas9-plus, SpCas9-HF1-plus, and B-SpCas9 are superior variants of eSpCas9, SpCas9-HF1, and WT SpCas9, respectively. The Blackjack (WT) SpCas9 provides higher fidelity editing than the WT without any detectable decrease in its on-target activity (Supplementary Fig. 2) and can be considered as an attractive general-purpose alternative to the WT enzyme for almost all applications. eSpCas9-plus and SpCas9-HF1-plus variants show identical fidelity and on-target activity to their original counterparts (eSpCas9 and SpCas9-HF1, respectively). Therefore, they would be excellent substitutes for eSpCas9 and SpCas9-HF1 in all types of applications for which the latter two have been preferentially used but with the added advantage of providing 20G-level editing with 21G-sgRNAs as well. This advantage is manifested when the sgRNA is transcribed from a DNA template (e.g., using a U6 or T7 promoter) and when the possibility of finding suitable sequences that can be targeted with 20G-sgRNAs

is limited. For example, if a specific position needs to be targeted by exploiting single-stranded oligonucleotides, such as when either dCas9-FokI nucleases or base editors are used or when tagging proteins. One of the most advantageous applications of the *plus* variants is the use of pooled sgRNA knockout libraries to decrease false-positive hits that frequently plague CRISPR screens. SpCas9-HF1-plus offers higher fidelity editing compared to B-SpCas9 or eSpCas9-plus, however, its activity on targets on average is decreased to 82% of that of the WT (Fig. 5c, e)[25,28]. Thus, its use would be profitable for applying to knockout libraries where a few sgRNAs are targeted to each gene, and the failure to cleave one out of every five target sites would be unlikely to spoil the experiment.

A 5′-GG extension of the sgRNA has been reported to increase the fidelity of WT SpCas9 and a similar effect is proposed for 21G-sgRNAs[28] as we show here (Fig. 3b). Since the use of 21G-sgRNAs does not alter much the target selectivity/on-target activity of WT SpCas9 (Supplementary Fig. 3a), 21G-sgRNAs could generally be employed instead of 20G-sgRNAs for almost all targets to provide higher specificity editing with WT SpCas9.

In conclusion, the incorporation of Blackjack mutations into an increased fidelity variant of SpCas9 that is not fully compatible with 5′ modified sgRNAs and using it either alone or in combination with a genome- or epigenome-editor would considerable increase its available target space by making them compatible with 21G-sgRNAs. Furthermore, Blackjack mutations may be combined with any SpCas9 or dSpCas9 editor or epigenome editor to increase editing specificity.

## Methods

**Materials**. Restriction enzymes, T4 ligase, Dulbecco's modified Eagle Medium DMEM (Gibco), fetal bovine serum (Gibco), Turbofect, Lipofectamine 2000, TranscriptAid T7 High Yield Transcription Kit, Qubit dsDNA HS Assay Kit, Shrimp Alkaline Phosphatase (SAP), Taq DNA polymerase (recombinant), Platinum Taq DNA polymerase, KnockOut Serum Replacement (Gibco), 0.45 μm sterile filters and penicillin/streptomycin were purchased from Thermo Fischer Scientific, protease inhibitor cocktail was purchased from Roche Diagnostics. DNA oligonucleotides, trimethoprim (TMP), chloroquine, polybrene, puromycin, calcium-phosphate and GenElute HP Plasmid Miniprep kit were acquired from Sigma-Aldrich. ZymoPure Plasmid Midiprep kit and RNA Clean & Concentrator kit were purchased from Zymo Research. NEBuilder HiFi DNA Assembly Master Mix and Q5 High-Fidelity DNA Polymerase were obtained from New England Biolabs Inc. NucleoSpin Gel and PCR Clean-up kit was purchased from Macherey-Nagel. Two millimeter electroporation cuvettes were acquired from Cell Projects Ltd, Bioruptor 0.5 ml Microtubes for DNA Shearing from Diagenode. Agencourt AMPure XP beads were purchased from Beckman Coulter. T4 DNA ligase (for GUIDE-seq) and end-repair mix were acquired from Enzymatics. KAPA universal qPCR Master Mix was purchased from KAPA Biosystems.

**Plasmid construction**. Vectors were constructed using standard molecular biology techniques including the one-pot cloning method[49–51], E. coli DH5α-mediated DNA assembly method[52], NEBuilder HiFi DNA Assembly and Body Double cloning method[53]. Plasmids were transformed into NEB Stable competent cells. For detailed cloning and sequence information see Supplementary Information. sgRNA target sites and mismatching sgRNAs sequences are available in Supplementary Data 1. The sequences of all plasmid constructs were confirmed by Sanger sequencing (Microsynth AG).

Plasmids acquired from the non-profit plasmid distribution service Addgene (http://www.addgene.org/) are the following:
pX330-U6-Chimeric_BB-CBh-hSpCas9 (Addgene #42230)[5], eSpCas9(1.1) (Addgene #71814)[24], VP12 (Addgene #72247)[25], sgRNA(MS2) cloning backbone (Plasmid #61424)[54], pMJ806 (#39312)[1], pBMN DHFR(DD)-YFP (#29325)[55] and p3s-Sniper-Cas9 (#113912)[45], LentiGuide-Puro (#52963)[56]. psPAX2 (#12260) and VSV-G envelope expressing plasmid (# 12259) are gifts from Didier Trono.

Plasmids developed by us and deposited at Addgene are the following:
pX330-Flag-dSpCas9 (Addgene #92113), pX330-Flag-WT_SpCas9 (without sgRNA; with silent mutations) (Addgene #126753), pX330-Flag-eSpCas9 (without sgRNA; with silent mutations) (Addgene #126754), pX330-Flag-SpCas9-HF1 (without sgRNA; with silent mutations) (Addgene #126755), pX330-Flag-HypaSpCas9 (without sgRNA; with silent mutations) (Addgene #126756),pX330-Flag-evoSpCas9 (without sgRNA; with silent mutations) (Addgene #126758), pX330-Flag-HeFSpCas9 (without sgRNA; with silent mutations) (Addgene #126759), pX330-Flag-Sniper SpCas9 (without sgRNA; with silent mutations)

(Addgene #126777), pX330-Flag-HiFi SpCas9 (without sgRNA; with silent mutations) (Addgene #126778),
B-SpCas9 (Addgene #126760), B-eSpCas9 (Addgene #126761), B-SpCas9-HF1 (Addgene #126762), B-HypaSpCas9 (Addgene #126763), B-evoSpCas9 (Addgene #126765), B-HeFSpCas9 (Addgene #126766)
eSpCas9-plus (Addgene #126767), SpCas9-HF1-plus (Addgene #126768)
pET-FLAG-eSpCas9 (Addgene #126769), pET-FLAG-SpCas9-HF1 (Addgene #126770), pET-FLAG-B-eSpCas9 (Addgene #126772), pET-FLAG-eSpCas9-plus (Addgene #126774), pET-FLAG-SpCas9-HF1-plus (Addgene #126775)
pmCherry_sgRNA-ver2 (Addgene #126776), pmCherry_gRNA (Addgene: #80457)

**Cell culturing**. Cells employed in the studies are N2a (neuro-2a mouse neuroblastoma cells, ATCC—CCL-131), HEK293 (Gibco 293-H cells), HEK293T (293T cells, ATCC—CRL-3216), N2a.dd-EGFP (a cell line developed by us containing a single integrated copy of an EGFP-DHFR[DD] [EGFP-folA dihydrofolate reductase destabilization domain] fusion protein coding cassette originating from a donor plasmid with 1000 bp-long homology arms to the *Prnp* gene driven by the *Prnp* promoter {*Prnp*.HA-EGFP-DHFR[DD]}) as well as N2a.EGFP and HEK-293. EGFP (both cell lines containing a single integrated copy of an EGFP cassette driven by the *Prnp* promoter)[28] cells. Cell lines were not authenticated as they were obtained directly from a certified repository or clone from those cell lines. Cells were grown at 37 °C in a humidified atmosphere of 5% $CO_2$ in high glucose Dulbecco's Modified Eagle medium (DMEM) supplemented with 10% heat inactivated fetal bovine serum, 4 mM L-glutamine (Gibco), 100 units/ml penicillin and 100 μg/ml streptomycin. Cells were passaged up to 20 times (washed with PBS, detached from the plate with 0.05% Trypsin-EDTA and replated). After 20 passages, cells were discarded.

**Flow cytometry**. Flow cytometry analyses were carried out on an Attune NxT Acoustic Focusing Cytometer (Applied Biosystems). For data collection and analysis Attune NxT Software v.2.7.0 was used. Viable single cells were gated based on side and forward light-scatter parameters and a total of 5000–10,000 viable single cell events were acquired in all experiments. The GFP fluorescence signal was detected using the 488 nm diode laser for excitation and the 530/30 nm filter for emission, the mCherry fluorescent signal was detected using the 488 nm diode laser for excitation and a 640LP filter for emission or using the 561 nm diode laser for excitation and a 620/15 nm filter for emission.

**EGFP-disruption assay**. All EGFP-disruption experiments were conducted on the N2a.dd-EGFP cell line except the on-target screen, which was conducted on N2a. EGFP cells (see details below). Cells were plated one day prior to transfection in 48-well plates at a density of approximately 25,000–30,000 cells/well. Cells were co-transfected with two types of plasmids: SpCas9 variant expression plasmid (137 ng) and sgRNA and mCherry coding plasmid (97 ng) using 1 μl TurboFect reagent per well in 48-well plates. TMP (trimethoprim; 1 μM final concentration) was added to the media ~48 h before FACS analysis in case of N2a.dd-EGFP cells. Transfected cells were analyzed ~96 h post-transfection by flow cytometry. Transfection efficacy was calculated via mCherry expressing cells. Transfections were performed in triplicate. Replicates not measured due to sample loss are indicated in the raw data (less than 1% in all experiments altogether).

Background EGFP loss for each experiment was determined using co-transfection of dead SpCas9 expression plasmid and different targeting sgRNA and mCherry coding plasmids. EGFP-disruption values were calculated as follows: the average EGFP background loss from dead SpCas9 control transfections made in the same experiment was subtracted from each individual treatment in that experiment and the mean values and the standard deviation (s.d.) were calculated from it. In the case of normalization, the results were normalized to the WT SpCas9 data from the same experiment.

On-target activity was measured on N2a.EGFP cell line 4 days post-transfection by flow cytometry. In this cell line the EGFP-disruption level is not saturated, this way this assay is a more sensitive reporter of the intrinsic activities of these nucleases compared to N2a.dd-EGFP cell line.

In the case of mismatch screens and 21G-sgRNA screens N2a.dd-EGFP cells were co-transfected with two types of plasmids: with SpCas9 variant expression plasmid (137 ng) and a mix of 3 sgRNAs in which one nucleotide position was mismatched to the target using all 3 possible bases and mCherry coding plasmid (3 × ~33.3 ng = 97 ng) using 1 μl TurboFect reagent per well in 48-well plates. TMP (trimethoprim; 1 μM final concentration) was added to the media ~48 h before FACS analysis. Transfected cells were analyzed ~96 h post-transfection by flow cytometry. The 4-day post-transfection results with this cell line show a close to saturated level, this way it is a good reporter system for seeing the full spectrum of off-target activities.

**Western blot**. N2a.dd-EGFP cells were cultured on 48-well plates and were transfected as described above in the EGFP-disruption assay section. Four days post-transfection, nine parallel samples corresponding to each type of SpCas9 variant transfected were washed with PBS, then trypsinized and mixed, and were analyzed for transfection efficiency via mCherry fluorescence level by using flow

cytometry. The cells from the mixtures were centrifuged at 200 rcf for 5 min at 4 °C. Pellets were resuspended in ice cold Harlow buffer (50 mM Hepes pH 7.5; 0.2 mM EDTA; 10 mM NaF; 0.5% NP40; 250 mM NaCl; Protease Inhibitor Cocktail 1:100; Calpain inhibitor 1:100; 1 mM DTT) and lysed for 20–30 min on ice. The cell lysates were centrifuged at 19,000 rcf for 10 min. The supernatants were transferred into new tubes and total protein concentrations were measured by the Bradford protein assay. Before SDS gel loading, samples were boiled in Protein Loading Dye for 10 min at 95 °C. Proteins were separated by SDS-PAGE using 7.5% polyacrylamide gels and were transferred to a PVDF membrane, using a wet blotting system (Bio-Rad). Membranes were blocked by 5% non-fat milk in Tris buffered saline with Tween20 (TBST) (blocking buffer) for 2 h. Blots were incubated with primary antibodies [anti-FLAG (F1804, Sigma) at 1:1000 dilution; anti-β-actin (A1978, Sigma) at 1:4000 dilution in blocking buffer] overnight at 4 °C. The next day after washing steps in TBST the membranes were incubated for 1 h with HRP-conjugated secondary anti-mouse antibody 1:20,000 (715-035-151, Jackson ImmunoResearch) in blocking buffer. The signal from detected proteins was visualized by ECL (Pierce ECL Western Blotting Substrate, Thermo Scientific) using a CCD camera (Bio-Rad ChemiDoc MP, Image Lab 4.1 Software).

**Indel analysis by next-generation sequencing (NGS)**. HEK293 cells were seeded onto 48-well plates a day before transfection at a density of $1.2 \times 10^4$ cells/well. The next day, at around 25% confluence, cells were transfected with plasmid constructs using Jetfect reagent (Biospiral-2006. Ltd.), briefly as follows: 234 ng total plasmid DNA (97 ng sgRNA and mCherry expression plasmid, and 137 ng nuclease expression plasmid) and 1 μl Jetfect reagent were mixed in 50 μl serum-free DMEM and the mixture was incubated for 30 min at room temperature prior adding to cells. Three parallel transfections were made from each sample. Replicates not measured due to sample loss are indicated in the raw data (less than 1%; Supplementary Data 2). Transfection efficiency was analyzed by flow cytometry 5 days post transfection via mCherry fluorescence after which cells were centrifuged at 1000 rcf for 10 min and genomic DNA was purified according to the Puregene DNA Purification protocol (Gentra systems). Amplicons for deep sequencing were generated using two rounds of PCR by Q5 high-fidelity polymerase to attach Illumina handles. The 1st step PCR primers used to amplify target genomic sequences are listed in Supplementary Data 1: PCR primers. After the 2nd step PCR the samples were quantified with Qubit dsDNA HS Assay kit and PCR products were pooled for deep sequencing. Sequencing on an Illumina Miseq instrument was performed by ATGandCo Ltd. Indels were counted computationally among reads that matched at least 75% to the first 20 bp of the reference amplicon. Indels without mismatches were searched at ±40 bp around the cut site. For each sample, the indel frequency was determined as (number of reads with an indel)/(number of total reads). Average reads per sample was 18,801 (see additional details in Supplementary Data 2). The following software was used: BBMap 38.08, samtools 1.8, BioPython 1.71, PySam 0.13. SRA accession PRJNA593843.

**In vitro transcription**. sgRNAs were in vitro transcribed using TranscriptAid T7 High Yield Transcription Kit and PCR-generated double-stranded DNA templates carrying a T7 promoter sequence. Primers used for the preparation of the DNA templates are listed in Supplementary Data 1: PCR primers. sgRNAs were dephosphorylated with SAP, purified with the RNA Clean & Concentrator kit, and reannealed (95 °C for 5 min, ramp to 4 °C at 0.3 °C/s). sgRNAs were quality checked using 10% denaturing polyacrylamide gels and ethidium bromide staining.

**Protein purification**. All SpCas9 variants were subcloned from pMJ806 (Addgene #39312)[1] (for detailed cloning information and sequence information see Methods: Plasmid construction section and Supplementary Information). The resulting fusion constructs contained an N-terminal hexahistidine (His6), a Maltose binding protein (MBP) tag and a Tobacco etch virus (TEV) protease site.

The expression constructs of the SpCas9 variants were transformed into *E. coli* BL21 Rosetta 2 (DE3) cells, grown in Luria-Bertani (LB) medium at 37 °C for 16 h. 10 ml from this culture was inoculated into 1 l of growth media (12 g/l Tripton, 24 g/l Yeast, 10 g/l NaCl, 883 mg/l NaH$_2$PO$_4$ H$_2$O, 4.77 g/l Na$_2$HPO$_4$, pH 7.5) and cells were grown at 37 °C to a final cell density of 0.6 OD600, and then were chilled at 18 °C. The protein was expressed at 18 °C for 16 h following induction with 0.2 mM IPTG. The protein was purified by a combination of chromatographic steps by NGC Scout Medium-Pressure Chromatography Systems (Bio-Rad). The bacterial cells were centrifuged at 6,000 rcf for 15 min at 4 °C. The cells were resuspended in 30 ml of Lysis Buffer (40 mM Tris pH 8.0, 500 mM NaCl, 20 mM imidazole, 1 mM TCEP) supplemented with Protease Inhibitor Cocktail (1 tablet/30 ml; complete, EDTA-free, Roche) and sonicated on ice. Lysate was cleared by centrifugation at 48,000 rcf for 40 min at 4 °C. Clarified lysate was bound to a 5 ml Mini Nuvia IMAC Ni-Charged column (Bio-Rad). The resin was washed extensively with a solution of 40 mM Tris pH 8.0, 500 mM NaCl, 20 mM imidazole, and the bound protein was eluted by a solution of 40 mM Tris pH 8.0, 250 mM imidazole, 150 mM NaCl, 1 mM TCEP. 10% glycerol was added to the eluted sample and the His6-MBP fusion protein was cleaved by TEV protease (3 h at 25 °C). The volume of the protein solution was made up to 100 ml with buffer (20 mM HEPES pH 7.5, 100 mM KCl, 1 mM DTT). The cleaved protein was purified on a 5 ml HiTrap SP HP cation exchange column (GE Healthcare) and

eluted with 1 M KCl, 20 mM HEPES pH 7.5, 1 mM DTT. The protein was further purified by size exclusion chromatography on a Superdex 200 10/300 GL column (GE Healthcare) in 20 mM HEPES pH 7.5, 200 mM KCl, 1 mM DTT and 10% glycerol. The eluted protein was confirmed by SDS-PAGE and Coomassie brilliant blue R-250 staining. The protein was stored at −20 °C.

**EGFP-disruption assay with RNP**. N2a.dd-EGFP cells cultured on 48-well plates, were seeded a day before transfection at a density of $3 \times 10^4$ cells/well, in 250 μl complete DMEM. 13.75 pmol SpCas9 and 16.5 pmol sgRNA was complexed in Cas9 storage buffer (20 mM HEPES pH 7.5, 200 mM KCl, 1 mM DTT and 10% glycerol) for 15 min at RT. 25 μl serum-free DMEM and 0.8 μl Lipofectamine 2000 was added to the complexed RNP and incubated for 20 min prior adding to the cells. TMP (trimethoprim; 1 μM final concentration) was added to the media ~48 h before FACS analysis. Transfected cells were analyzed ~96 h post-transfection by flow cytometry. Transfections were performed in triplicate. Background EGFP loss for each experiment was determined using co-transfection of WT SpCas9 expression plasmid and non-targeted sgRNA and mCherry coding plasmids. EGFP-disruption values were calculated as follows: the average EGFP background loss from control transfections made in the same experiment was subtracted from each individual treatment in that experiment and the mean values and the standard deviation (s.d.) were calculated from it.

**GUIDE-seq**. GUIDE-seq experiments were performed with WT SpCas9, B-SpCas9, eSpCas9, eSpCas9-*plus*, SpCas9-HF1, SpCas9-HF1-*plus* and B-eSpCas9 on thirteen different target sites. Briefly, $2 \times 10^6$ HEK293.EGFP cells were transfected with 3 μg of SpCas9 variant expressing plasmid, 1.5 μg of mCherry and sgRNA coding plasmid. 100 pmol of the dsODN containing phosphorothioate bonds at both ends (according to the original GUIDE-seq protocol[23]) was mixed together with 100 μl home-made nucleofection solution to the plasmid and electroporated as described in Vriend et al.[57] using Nucleofector (Lonza) with A23 program and 2 mm electroporation cuvettes.

Transfected cells were analyzed 3 days post-transfection by flow cytometry. Cells were then centrifuged at 1000 rcf for 10 min and genomic DNA was purified according to Puregene DNA Purification protocol (Gentra systems). Genomic DNA was sheared with BioraptorPlus (Diagenode) to 550 bp in average. Sample libraries were assembled as previously described[23] and sequenced on an Illumina MiSeq instrument by ATGandCo Ltd. Data were analyzed using open-source guideseq software (version 1.1)[58]. Consolidated reads were mapped to the human reference genome GrCh37 supplemented with the integrated EGFP sequence. Upon identification of the genomic regions integrating double-stranded oligodeoxynucleotide (dsODNs) in aligned data, off-target sites were retained if at most seven mismatches against the target were present and if absent in the background controls. Visualization of aligned off-target sites are provided as a color-coded sequence grid. Further details can be found in Supplementary Data 3 and GUIDE-seq sequencing data are deposited at NCBI Sequence Read Archive: PRJNA593843.

**TIDE**. Tracking of Indels by DEcomposition (TIDE) method[59] was applied for analyzing mutations and determining their frequency in a cell population using different sgRNAs and SpCas9 proteins. From the isolated genomic DNA PCR was conducted with Q5 High-Fidelity DNA Polymerase in triplicates (for PCR primer details, see Supplementary Data 1). Genomic PCR products were gel excised via NucleoSpin Gel and PCR Clean-up kit and were Sanger sequenced. Indel efficiencies were analyzed by TIDE webtool (https://tide.nki.nl/) by comparing SpCas9 treated and control samples.

**HR mediated pre-screening of the Shadoo gene target sites**. N2a cells were seeded into 48-well plates a day before transfection at a density of $2.5 \times 10^4$ cells/well. Next day cells were co-transfected with three types of plasmids: an expression plasmid for EGFP flanked by 1000 bp-long homology arms to the Shadoo (*Sprn*) gene (*Sprn*.HA-CMV-EGFP plasmid) (166 ng), SpCas9 expressing plasmid (42 ng) and an sgRNA/mCherry coding plasmid (42 ng), giving 250 ng total plasmid DNA, using 1 μl TurboFect reagent per well. Transfected cells were analyzed 4- and 18-days post-transfection by flow cytometry. Transfection efficiency was calculated via mCherry expressing cells measured 4 days post-transfection. EGFP positive cells were counted 18 days post-transfection. Transfections were performed in triplicate.

**NHEJ-mediated integration using a 'self-cleaving' plasmid[48]**. N2a cells were seeded into 12-well plates a day before transfection at a density of $8 \times 10^4$ cells/well. Next day cells were co-transfected with three types of plasmids: a 'self-cleaving' EGFP-expression plasmid[48] (which has to integrate in-frame for *Sprn* promoter driven EGFP expression; for sequence details see Supplementary Information) (1 μg), SpCas9 expressing plasmid (590 ng) and an sgRNA/mCherry coding plasmid (410 ng), giving 2 μg total plasmid DNA, using 4 μl TurboFect reagent per well. Transfections were performed in triplicate. Transfection efficiency was calculated via mCherry expressing cells measured 4 days post-transfection. EGFP positive cells were counted 14 days post-transfection.

**Lentivirus production and transduction.** For the lentiviral approach we cloned the DNA oligos into the lentiGuide-puro 3rd generation lentiviral transfer vector between BsmBI restriction sites. sgRNA target sites are available in Supplementary Data 1. The lentiGuide-puro transfer vectors were co-transfected with a 2nd generation lentiviral packaging plasmid (psPAX2) and a VSV-G envelope expressing plasmid (pMD2.G) using calcium-phosphate transfection method in HEK293T cells, briefly: HEK293T cells were seeded onto 6-well plates in $5.7 \times 10^5$ cell/well density. Twenty-four hours after seeding the medium volume on cells was reduced to 1 ml and cells were treated with chloroquine (25 μM final concentration). Transfection mixtures were prepared as follows: 1.35 μg psPAX2, 0.75 μg pMD2.G, 25 μg sgRNA expressing transfer vector, 8.055 μl 2.5 M $CaCl_2$ were mixed in 705 μl sterile distilled water. After 5 min RT incubation 705 μl 2× HEBS buffer (50 mM HEPES, 280 mM NaCl, 1.5 mM, $Na_2HPO_4$, pH adjusted to 7.0, sterile filtered) was added drop-wise while the solution was mixed vigorously and then the whole mixture was added slowly to the cells. Transfection medium was changed to virus-gathering medium (DMEM complemented with KnockOut Serum Replacement) 18 h post-transfection. Lentivirus was collected 48 h post-transfection, filtered through 0.45 μm sterile filters to remove debris, aliquoted and stored at −80 °C.

Lentivirus titer was calculated in HEK293T cells after 72 h of puromycin (2 ug/ml) selection. Cells were trypsinized and the viable cells were counted after trypan blue staining using TC20 Automated Cell Counter (BioRad). Virustiter was calculated using the following formula: 60,000/Vvirus × cell%, where Vvirus is the volume of the virus used for transduction in ml, cell% is the percentage of the surviving cells (number of surviving cells after puromycin treatment/number of surviving cells without puromycin treatment × 100%).

To generate stable sgRNA expressing cell cultures HEK293T cells were transduced with the sgRNA encoding lentiviruses in the MOI range 0.1–0.5, in the presence of 6 μg/ml polybrene. Forty-eight hours post-transduction cells were passaged and were treated with puromycin (2 μg/ml) for 6 days before the surviving populations were expanded and used to test the SpCas9 variants' activities.

All sgRNA expressing cell lines were plated one day prior to transfection in 48-well plates at a density of approximately 30,000 cells/well. Cells were co-transfected with two types of plasmids: SpCas9 variant expression plasmid (137 ng) and mCherry coding plasmid (97 ng) using 1 μl TurboFect reagent per well in 48-well plates. Transfected cells were analyzed 5 days post-transfection by flow cytometry. Transfection efficacy was calculated via mCherry expressing cells. Transfections were performed in triplicate. TIDE method was applied for analyzing mutations and determining their frequency.

**Statistics.** Differences between SpCas9 variants were tested by using either Paired-samples Student's *t*-test (Fig. 2a: WT SpCas9/B-SpCas9, eSpCas9/B-eSpCas9; Fig. 2b: evoSpCas9/B-evoSpCas9; Fig. 3a: evoSpCas9/B-evoSpCas9, HeFSpCas9/B-HeFSpCas9; Supplementary Fig. 8f: eSpCas9-ribosyme/eSpCas9-tRNA) or by using Wilcoxon Signed Ranks test(Fig. 2a: SpCas9-HF1/B- SpCas9-HF1, HypaSpCas9/B-HypaSpCas9, evoSpCas9/B-evoSpCas9, HeFSpCas9/B-HeFSpCas9; Fig. 2b: eSp-Cas9/B-eSpCas9, SpCas9-HF1/B- SpCas9-HF1, HypaSpCas9/B-HypaSpCas9; Fig. 3a: eSpCas9/B-eSpCas9, SpCas9-HF1/B- SpCas9-HF1, HypaSpCas9/B-HypaSpCas9; Supplementary Fig. 4b: WT SpCas9/B-SpCas9; Supplementary Fig. 8f: eSpCas9/eSpCas9-*plus*, eSpCas9/eSpCas9-ribosyme, eSpCas9/eSpCas9-tRNA, eSpCas9-*plus*/eSpCas9-ribosyme, eSpCas9-*plus*/eSpCas9-tRNA) in cases where differences did not meet the assumptions of Paired *t*-test. Normality of data and of differences were tested by Shapiro-Wilk normality test. Statistical tests were performed using IBM SPSS ver. 20 on data including all parallel sample points. Test results are shown in Supplementary Data 6.

**Reporting summary.** Further information on research design is available in the Nature Research Reporting Summary linked to this article.

## Data availability

Important plasmids used in this study are available at Addgene (details in Plasmid Construction section). Sequences of the constructs are listed in Supplementary information. All associated raw data are available in Supplementary Data 4 and 5 for Figs. 1–7 and Supplementary Fig. 1–8, respectively; Supplementary Data 3 for Figs. 3b, 5f and Supplementary Figs. 5 and 7. The data that support the findings of this study are available from the corresponding authors upon request as well. The deep-sequencing data (targeted deep-sequencing and GUIDE-seq) from this study have been submitted to the NCBI Sequence Read Archive (SRA; http://www.ncbi.nlm.nih.gov/sra/) under accession number: PRJNA593843.

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

## Acknowledgements

We thank Ildikó Szűcsné Pulinka, Judit Szűcs, Bernadett Czene, Dávid Fetter, Gábor Erdős, Orsolya Oravecz, Balázs Bohár for their excellent laboratory assistance, Lőrinc Pongor and Shengdar Tsai for providing scripts, Elfrieda Fodor, Edit Szabó, György Várady, Tamás Hegedűs for their valuable help. This research was supported by the National Research, Development and Innovation Office [K128188 to E.W. and VKFI-16-1-2016-0240 to Z.L.] and by the Ministry of National Economy [GINOP-2.1.7-15-2016-00584].

## Author contributions

P.I.K. and E.W. conceived and designed experiments, P.I.K., A.T., E.T., A.N., and Z. L. performed all experiments. P.I.K., A.T., E.T., A.N., Z.W., and Z. L. analyzed the data. P.I.K. and E.W. wrote the manuscript with input from all the authors.

## Competing interests

Biospirál-2006 Ltd has filed a patent (Application number 128977–18457) concerning the use of Blackjack variants described in this manuscript with P.I.K., A.T., E.T., A.N., Z.L., Z.W. and E.W. as listed inventors.
