## [Peer Review File · Nature Communications]

Reviewers' Comments:

Reviewer #1:

Remarks to the Author:

The authors have addressed some of the referees' critiques and on balance I think the manuscript contributes valuable data regarding the accuracy and efficiency of Cas-mediated genome editing using guide RNAs expressed from U6 or T7 promoters. Reviewer #2 in particular raises valid points about the specialized focus of the study and the lack of clarity of presentation. I support publication of the revised manuscript despite these limitations.

Reviewer #3:

Remarks to the Author:

The revised version of this manuscript describes 'blackjack' variants of high-fidelity *S. pyogenes* Cas9 nucleases that rescue the activity of these nucleases when associated with a guideRNA (gRNA) that contains an additional 5' G. Initially, a number of high-fidelity SpCas9 variants are characterized for activity and specificity in an EGFP-reporter assay. Next, mutants of SpCas9-HF1 are identified that rescue its activity when used with U6-promoter driven gRNAs. Some points are clarified, such as the comparison of blackjack variants with other gRNA processing methods.

In general, this version of the manuscript trades clarity for comprehensiveness and subpanels of the main figures do not always contribute to a single cohesive point. I would encourage the authors to thoughtfully edit this manuscript with the consideration that sometimes less is more, and that presenting a few clear well-defined experiments may be superior to presenting many more ambiguous ones. Otherwise, the argument that blackjack mutations may be useful for rescuing the activity of high-fidelity SpCas9 nucleases in certain contexts (like pooled screening) is easily lost.

I would suggest that the authors show clearly a story in the format of: 1) optimization of blackjack variants, 2) blackjack variants rescue activity of high-fidelity variants with gRNAs that contain a single 5' G extension in EGFP reporter, 3) Blackjack variants rescue activity of high-fidelity variants at endogenous genes, 4) effect of blackjack mutations on genome-wide activity, and 5) Blackjack variants expand useful activity of high-fidelity variants in lentiviral expression for pooled screening.

Main points:

1. These blackjack mutations are only useful in applications that require DNA-based encoding of Cas9 such as genome-scale library screening. Otherwise, there is no requirement for a 5' G for Cas9 when delivered as an RNP. If pooled library screens are indeed the main application for this variant, then a demonstration that they work in this context is critical.
2. The effect of blackjack mutations on activity at endogenous genes should be clearly presented. For example, the data of 4f should be broken out by targets and at least 10-20 targets should be tested.
3. The main message of figure 1 remains unclear. Certain high-fidelity enzymes do not have on-target activity at a large proportion of sites. Some of the enzymes that have lower on-target activities have better discrimination at sites where they are active. Only point mutations are tested.
4. In figure 3c, arguments for activity are clear, but data supporting specificity are limited for two

reasons: 1) they are based on tests of a single mismatch at one position, and 2) they are based on an artificial scenario where mismatched gRNAs are tested against a fixed target, rather than a fixed gRNA against mismatched off-target sites.

5. For many of the experiments in the manuscript, it would be clearer to organize the plots by variant rather than target site. For example, for Supplementary Fig. 1d-h, the activity for all the targets tested for HF-B9 (B-SpCas9-HF1) grouped together would make it easier to interpret the data. In the case of B-SpCas9-HF1, it appears that activity is sometimes compromised with the blackjack mutations and that rescue of activity when adding an additional 5'G is inconsistent.

6. How do the authors suggest that potential users apply blackjack variants? It is impractical to suggest that users screen an entire matrix of high-fidelity blackjack variants against potential targets to identify a single nuclease and target combination that is the best. Is there a blackjack variant that is most generally useful?

> Reviewer #3 (Remarks to the Author):

>

> The revised version of this manuscript describes ‘blackjack’
> variants of high-fidelity *S. pyogenes* Cas9 nucleases that rescue the
> activity of these nucleases when associated with a guideRNA (gRNA) that
> contains an additional 5’ G. Initially, a number of high-fidelity
> SpCas9 variants are characterized for activity and specificity in an
> EGFP-reporter assay. Next, mutants of SpCas9-HF1 are identified that
> rescue its activity when used with U6-promoter driven gRNAs. Some points
> are clarified, such as the comparison of blackjack variants with other
> gRNA processing methods.

>

> In general, this version of the manuscript trades clarity for
> comprehensiveness and subpanels of the main figures do not always
> contribute to a single cohesive point. I would encourage the authors to
> thoughtfully edit this manuscript with the consideration that sometimes
> less is more, and that presenting a few clear well-defined experiments may
> be superior to presenting many more ambiguous ones. Otherwise, the
> argument that blackjack mutations may be useful for rescuing the activity
> of high-fidelity SpCas9 nucleases in certain contexts (like pooled
> screening) is easily lost.

>

> I would suggest that the authors show clearly a story in the format of: 1)
> optimization of blackjack variants, 2) blackjack variants rescue activity
> of high-fidelity variants with gRNAs that contain a single 5’ G
> extension in EGFP reporter, 3) Blackjack variants rescue activity of
> high-fidelity variants at endogenous genes, 4) effect of blackjack
> mutations on genome-wide activity, and 5) Blackjack variants expand useful
> activity of high-fidelity variants in lentiviral expression for pooled
> screening.

Agreeing with the comments of the reviewer we reorganized the manuscript to show our results in the suggested order except that we reversed the order of points 3 and 4 to accommodate the development of the *plus* variants. We have revised the results section of the manuscript accordingly. >

> Main points:

> 1. These blackjack mutations are only useful in applications that require
> DNA-based encoding of Cas9 such as genome-scale library screening.
> Otherwise, there is no requirement for a 5’ G for Cas9 when
> delivered as an RNP. If pooled library screens are indeed the main
> application for this variant, then a demonstration that they work in this
> context is critical.

We have answered this comment to ensure that one of the most important applications of the Blackjack variants does not get lost in an overly complex manuscript, about which the reviewer warned (see above), and we examined the activity of eSpCas9-*plus* compared to WT and eSpCas9 employing four sgRNAs transcribed from an integrated single copy of lentivirus as shown in the new Fig. 7c. eSpCas9-*plus*, in contrast to eSpCas9, demonstrated activities approaching that of the WT protein. These results confirm that the activity of eSpCas9-*plus* is compatible with pooled sgRNA libraries where the sgRNAs are naturally expressed from an integrated single copy of lentivirus and even when 21G-sgRNAs are used. We would also be interested in investigating a pooled sgRNA library KO screen performed with *plus* variants. However, to adjust the conditions of a KO pooled sgRNA screen and to finish these experiments including its evaluation

would require from 3 to 6 months. Considering that our manuscript had been in review for four months in Nature Cell biology and for more than another three months in Nature Communication, we feel a further delay would compromise its timeliness much more than the benefits that would arise from including these lentiviral experiments. We described the results on page 13 lines 272-278 and in new Fig. 7c.

- > 2. The effect of blackjack mutations on activity at endogenous genes
- > should be clearly presented. For example, the data of 4f should be broken
- > out by targets and at least 10-20 targets should be tested.

The activity of two Blackjack variants (eSpCas9-plus and SpCas9-HF1-plus) on 23 endogenous targets is presented in the new Figures 5d and 5e. The data are broken out by 20G-19N-NGG and 21G-20N-NGG target types, respectively. The data are also broken out by single targets in our new Supplementary Figure 6g.

- > 3. The main message of figure 1 remains unclear. Certain high-fidelity
- > enzymes do not have on-target activity at a large proportion of sites.
- > Some of the enzymes that have lower on-target activities have better
- > discrimination at sites where they are active. Only point mutations are
- > tested.

Following the advice of the reviewer and since the main messages of the old Figure 1 do not necessarily help the story be more easily understood we deleted it from the MS.

- > 4. In figure 3c, arguments for activity are clear, but data supporting
- > specificity are limited for two reasons: 1) they are based on tests of a
- > single mismatch at one position, and 2)
- > they are based on an artificial scenario where mismatched gRNAs are tested
- > against a fixed target, rather than a fixed gRNA against mismatched
- > off-target sites.

High fidelity nucleases generally have a strongly decreased off-target propensity, while they cleave many targets with a single mismatch, their activity with two or more mismatches is strongly reduced. To see the effect of the Blackjack mutations, we needed to use a scenario where the fidelity increasing effect of the Blackjack mutations compared to the other variants could be clearly detected. That is why we used single mismatches. Actually, we screened the effect of mismatches on three positions in the case of 16 target sites (on six PAM distal positions), using the mix of three mismatching guides for each position (altogether 144 mismatching sgRNAs). We think it is a sufficiently high number to support our conclusion.

We also agree that this mismatch screen is an artificial scenario and that the mismatching sgRNAs may not provide complete information about how much off-targets would be cleaved when a particular target sequence is edited with these nucleases. However, using the same mismatching target/sgRNA pairs to compare the mismatch-tolerance of the SpCas9 nucleases provides a useful guide. We also agree that a mismatch screen does not provide as comprehensive a picture of the off-target propensity of the nucleases that a genome-wide unbiased method, such as GUIDE-seq, is capable of providing. This is why, for the characterization of the off-target propensity of the three most important Blackjack variants (B-SpCas9, eSpCas9-plus and SpCas9HF1-plus) along with their parental variants, we have also included GUIDE-seq (new Figure 3b and 5f and new Supplementary Figure 5 and 7, in the revised MS) in this version of the MS.

> 5. For many of the experiments in the manuscript, it would be clearer to
> organize the plots by variant rather than target site. For example, for
> Supplementary Fig. 1d-h, the activity for all the targets tested for HF-B9
> (B-SpCas9-HF1) grouped together would make it easier to interpret the
> data. In the case of B-SpCas9-HF1, it appears that activity is sometimes
> compromised with the blackjack mutations and that rescue of activity when
> adding an additional 5'G is inconsistent.

We reorganized Supplementary Figures 2 and 3 according of the advice of the reviewer and confirm that the data is more understandable when presented in this way. We also revised Supplementary Figure 1 but the nature of the data prevented a comparable reorganization. These panels represent separate experiments. Although comparing the various variants on these targets gives reproducible results, the absolute values of the data deriving from separate experiments differ, which prevents them being organized into a single panel. The reviewer is also right to point out that the Blackjack mutations sometimes compromise the activity of the increased fidelity nucleases with 20G- or 21G-sgRNAs as exemplified by SpCas9-HF1. It is due to their fidelity increasing effect - an effect that we confirmed both on the WT protein (new Figure 3b) and on some of the increased fidelity variants (new Figure 3a). It is also the reason why we developed the *plus* variants, which show no compromised activity on targets with 20- or 21G-sgRNAs when compared to their non-Blackjack original counterparts.

> 6. How do the authors suggest that potential users apply blackjack
> variants? It is impractical to suggest that users screen an entire matrix
> of high-fidelity blackjack variants against potential targets to identify
> a single nuclease and target combination that is the best. Is there a
> blackjack variant that is most generally useful?

We highlighted three Blackjack variants for general use: B-SpCas9, eSpCas9-plus and SpCas9-HF1-plus that are superior to the WT SpCas9, eSpCas9 and SpCas9-HF1, respectively. Blackjack WT SpCas9 (B-SpCas9) has identical on-target activity but increased fidelity compared to WT SpCas9. When using 20G-sgRNAs, eSpCas9-plus and SpCas9-HF1-plus have identical fidelity and on-target activity compared to eSpCas9 and SpCas9-HF1, respectively. However, they have increased on-target activity with 21G-sgRNAs. For selecting the most suitable variants, users can apply the same rationale as that used for to decide when to apply the WT SpCas9, the eSpCas9 or the SpCas9-HF1 nucleases. To highlight the usefulness of these variants among the other Blackjack nucleases we included a discussion in page 14 lines 281-299 as the reviewer suggested.

We are grateful for this reviewer for the time he or she has devoted to help perfect our manuscript. We feel that it has been considerably improved thanks to his/her advice.

Reviewers' Comments:

Reviewer #3:

Remarks to the Author:

The authors have done a nice job in reorganizing the presentation of the manuscript for clarity and have addressed my major concerns. The addition of additional RNP data and tests in a lentiviral expression context have additionally strengthened the paper, making the revised manuscript suitable for publication in Nature Communications. Some minor editing suggestions are listed below.

Line 117. There is an extra comma.

Figure 2 legend should be edited to remove reference to target selectivity since this is now only addressed in Figure 3.

It may be worth mentioning that the Vakulsas et al. study performed many of their tests on endogenous gene targets including HPRT which may behave differently because of chromatin context than the EGFP-disruption tests.

Finally, I would suggest generalizing the conclusion to discuss applications of Blackjack mutation to any Cas9 or dCas9 genome and epigenome editors, rather than focusing so specifically on xCas9.

The reviewer's comments:

Line 117. There is an extra comma.

Removed.

Figure 2 legend should be edited to remove reference to target selectivity since this is now only addressed in Figure 3.

Although the main message of the figure is about the activity of the variants with 21G-sgRNA, the figure also shows the increase of the target selectivity of the variants in general. We clarified this in the figure title (page 34. lines 772-773.).

It may be worth mentioning that the Vakulsas et al. study performed many of their tests on endogenous gene targets including HPRT which may behave differently because of chromatin context than the EGFP-disruption tests.

We included a sentence on (page 12. lines 250-253.) to indicate that it is one of the differences between the two study.

Finally, I would suggest generalizing the conclusion to discuss applications of Blackjack mutation to any Cas9 or dCas9 genome and epigenome editors, rather than focusing so specifically on xCas9.

We agreed with the reviewer on this point and revised the last few sentences of the discussion according to his/her advice on (page 15. lines 310-314.).